# Identification of Multiple Hub Genes in Acute Kidney Injury after Kidney Transplantation by Bioinformatics Analysis

**DOI:** 10.3390/medicina58050681

**Published:** 2022-05-20

**Authors:** Sang-Wook Kang, Sung-Wook Kang, Ju-Yeon Ban, Min-Su Park

**Affiliations:** 1Department of Oral and Maxillofacial Pathology, School of Dentistry, Kyung Hee University, Seoul 02447, Korea; ifthisplus88@hanmail.net; 2Neuroscience Center of Excellence, Louisiana State University School of Medicine, New Orleans, LA 70112, USA; khksw0403@gmail.com; 3Department of Dental Pharmacology, School of Dentistry, Dankook University, Cheonan 31116, Korea; 4Department of Surgery, School of Medicine, Kyung Hee University, Seoul 02447, Korea

**Keywords:** acute kidney injury, bioinformatics analysis, hub genes, kidney transplantation, pathway

## Abstract

*Background and Objectives*: The molecular mechanisms of the development of acute kidney injury (AKI) after kidney transplantation are not yet clear. The aim of this study was to confirm the genes and mechanisms related to AKI after transplantation. *Materials and Methods*: To investigate potential genetic targets for AKI, an analysis of the gene expression omnibus database was used to identify key genes and pathways. After identification of differentially expressed genes, Kyoto Encyclopedia of Genes and Genome pathway enrichment analyses were performed. We identified the hub genes and established the protein–protein interaction network. *Results*: Finally, we identified 137 differentially expressed genes (59 upregulated genes and 16 downregulated genes). AKAP12, AMOT, C3AR1, LY96, PIK3AP1, PLCD4, PLCG2, TENM2, TLR2, and TSPAN5 were filtrated by the hub genes related to the development of post-transplant AKI from the Protein–Protein Interaction (PPI) network. *Conclusions*: This may provide important evidence of the diagnostic and therapeutic biomarker of AKI.

## 1. Introduction

Kidney transplantation is the effective treatment for end-stage kidney disease, and kidney transplant survival rates have improved significantly due to immunological developments and the advent of powerful immunosuppressants [1,2]. However, renal dysfunction in the early stage after transplantation has a known profound effect on long-term patient prognosis. Therefore, to improve the graft survival rate, it is very important to quickly diagnose and appropriately treat early renal dysfunction.

Delayed graft function (DGF), a condition requiring dialysis within a week after kidney transplantation, is a sign of acute kidney injury (AKI) that occurs during transplantation [3]. DGF reportedly occurs at an 8–50% rate after kidney transplantation and can affect the length of hospitalization, hospitalization costs, quality of life during hospitalization, dialysis-related complications, and social and professional rehabilitation progress. DGF is generally known to increase the acute rejection rates and cause fibrosis of the transplanted kidney that decreases its function [4,5].

Old donors, prolonged ischemic time, high donor serum creatinine levels, and non-heartbeat donation contribute to AKI development after kidney transplantation. Multiple genetic and molecular pathways are presumed to be associated with the development of post-transplant AKI; however, the genetic mechanisms remain unclear [6,7,8]. To enable the adequate diagnosis of and therapy for AKI, the identification of the novel related biomarkers is crucial.

Bioinformatics analysis has recently been widely applied to investigate the mechanisms of various diseases. However, the molecular mechanisms and effects of AKI after kidney transplantation have not been examined much. Analysis of whole-genomic mRNA and miRNA expression profiles in kidney transplant patients can play an important role in studying the development of AKI after transplantation.

Therefore, in this study, we executed an integrative bioinformatics analysis to identify hub genes associated with AKI after transplantation, using three data sets with mRNA and miRNA expression information.

## 2. Methods

### 2.1. Data Source

In this study, we analyzed datasets from the Gene Expression Omnibus (GEO) database [http://www.ncbi.nlm.nih.gov/geo (accessed on 2 January 2022)] related to AKI after kidney transplantation. In the Prue AKI dataset, acute rejection and renal disease by histologic criteria, and non-diagnostic histological lesions were excluded. We selected the “pure AKI” datasets with DGF. In the end, we selected three separate gene expression profiles (GSE37838, GSE30718, and GSE53769) based on the GPL570 platform.

### 2.2. Differentially Expressed Genes (DEGs) Analyses

Data normalization and differential expression analyses were performed using the R packages affy and limma, and the GEO2R online tool in NCBI [http://www.ncbi.nlm.nih.gov/geo/geo2r (accessed on 20 January 2022)]. The cutoff criteria for defining DEG were adjusted *p* values < 0.05 and absolute log2FC values > 2.

### 2.3. Gene Ontology (GO) and Kyoto Encyclopedia of Genes and Genome (KEGG) Pathway Analysis of DEGs

The Database for Annotation, Visualization and Integrated Discovery [DAVID; http://david.ncifcrf.gov (accessed on 25 January 2022)] tools were used. Gene Ontology (GO) annotation and Kyoto Encyclopedia of Genes and Genomes (KEGG) pathway enrichment analyses were executed. Values of *p* < 0.05 were considered statistically significant.

### 2.4. Integration of the Protein–Protein Interaction (PPI) Network and Hub Gene Identification

The PPI network analysis using the Search Tool for the Retrieval of Interacting Genes (STRING) database [http://www.string-db.org (accessed on 25 January 2022)] was carried out in order to obtain the connections between proteins encoded by DEGs and significant gene modules involved in AKI development after kidney transplantation. We visualized a PPI network with a confidence score ≥ 0.4 using Cytoscape (3.8.0).

## 3. Results

### 3.1. Identification of DEGs

Here, we selected three groups of gene expression profiles (GSE37838, GSE30718, and GSE53769). GSE37838 contained 12 AKI and 58 normal specimens (all deceased donors), GSE30718 contained 28 AKI and 11 normal specimens (24 deceased donors and 13 living donors), and GSE53769 included 17 AKI and 17 normal samples (all deceased donors). After 8825 DEGs were identified, we performed a Venn analysis. Overall, 137 DEGs were identified (59 upregulated genes and 16 downregulated genes) (Figure 1).

### 3.2. Functional Enrichment Analyses of the DEGs

The DEG biological functions in AKIs, GO function, and KEGG pathway enrichment analyses were performed by DAVID (Figure 2). Cellular component (CC), biological process (BP), and molecular function (MF) ontologies were analyzed. The DEGs were mainly enriched in secretory granule membrane, secretory granule, cell surface, transcription factor complex, and chromatin in the CC analysis. The DEGs were significantly enriched in sequence-specific double-stranded DNA binding, histone deacetylase binding, and neurexin family protein binding in the MF analysis. The DEGs were significantly enriched in negative regulation of the cholangiocyte apoptotic process and cellular response to transforming growth factor-beta stimulus and in positive regulation of cell proliferation in the BP analysis. In addition, terms associated with proteoglycan in cancer, transcriptional mis-regulation in cancer, and pathways in cancer were obtained from the KEGG pathway enrichment analysis results.

### 3.3. PPI Network Construction and Hub Gene Identification

As constructed based on the STRING online tool by Cytoscape 3.8.0 (https://cytoscape.org/ (accessed on 25 January 2022), Cytoscape Consortium, USA), the PPI network of the 137 identified DEGs was identified (Figure 3). Furthermore, in this network, the top 10 genes with the highest connectivity were selected as hub genes (Table 1 and Figure 4), including A-kinase anchoring protein 12 (AKAP12), angiomotin (AMOT), complement C3a receptor 1 (C3AR1), lymphocyte antigen 96 (LY96), phosphoinositide-3-kinase adaptor protein 1 (PIK3AP1), phospholipase C delta 4 (PLCD4), phospholipase C gamma 2 (PLCG2), teneurin transmembrane protein 2 (TENM2), Toll-like receptor 2 (TLR2), and tetraspanin 5 (TSPAN5).

## 4. Discussion

AKI is emerging as a major problem after kidney transplantation with inferior allograft outcomes. Several donor and recipient risk factors, along with cold and warm ischemia, exacerbate the development of AKI. Oxidative stress, cytokine signaling, vasospasm, innate immunity, adaptive immunity, endothelial cell injury, and epithelial cell injury play important roles in the pathogenesis of AKI [3,9,10,11]. However, the exact molecular mechanism of AKI after transplantation is unknown.

The identification of new hub genes and molecular pathways involved in the development of AKI after kidney transplantation may aid in the early diagnosis and treatment of AKI. Here, we used bioinformatics to analyze gene expression datasets for GSE37838, GSE30718, and GSE53769 and screen for hub genes associated with the AKI development. 

We used gene expression and PPI analysis to identify the hub genes associated with post-transplant AKI. We found 137 DEGs in AKI versus normal tissue by comparing gene expression profiling data from the three GEO datasets. Next, we built the PPI network and identified 10 hub genes, including AKAP12, AMOT, C3AR1, LY96, PIK3AP1, PLCD4, PLCG2, TENM2, TLR2, and TSPAN5. AKAP12 is related to protein kinases A and C and phosphatase and functions as a scaffold protein. A recent study found that AKAP12 was associated with renal epithelial cell function and renal tubule morphogenesis [12]. AMOT interacts with Yes-related proteins to inhibit or stimulate protein activity and play an important role in cell proliferation. AMOT promoted the proliferation of renal epithelial cells and enhances the progression of renal cell carcinoma [13]. AMOT reduced glomerular hypertrophy and protected against diabetic nephropathy as a novel angiogenesis modulator [14]. 

C3AR1 reduces renal inflammation and preserves renal function. It also contributes to the pathogenesis of renal ischemia–reperfusion injury [15]. C3AR1 is crucial for the protective factor and an independent risk factor for renal cell carcinoma. LY96 is essential to the proliferation of kidney inflammation in chronic renal disease and closely related to proliferation of renal cell carcinoma with a poor prognosis [16,17]. PIK3AP1 regulates the inflammatory signal transduction and Toll-like receptor signal transduction. Upregulated expressed genes of PIK3AP1 were associated with antibody-mediated rejection in kidney transplantation in the analysis of GEO datasets [18]. 

The phospholipase C gene encoded by PLCD4 is crucial for the functional development of the glomeruli and the development of nephrotic syndrome [19]. PLCG2 is closely associated with auto-inflammation, antibody deficiency, and immune dysregulation. It plays a major role in adaptive and autoimmunity in the pathogenesis of nephrotic syndrome [20]. TENM2, which enables signaling receptor binding activity and cell adhesion molecule binding activity, is involved with renal fibrosis and the development of diabetic kidney disease [21]. TLR2, which plays a fundamental role in pathogen recognition and innate immunity activation, is essential to the pro-inflammatory and detrimental role in the kidney after ischemic reperfusion injury [22]. Overactivated TLR2 with cell apoptosis was related to septic AKI in an animal model [23]. TSPAN5 protein modulates signal transduction in the regulation of the development, growth, and activation of the cell. TSPAN5 is known to regulate the inflammatory response by interacting with adam10 in the kidney and causes kidney disease [24].

Several studies have recently been reported showing the bioinformatic analysis of AKI development after transplantation [25,26]. They examined different GEO datasets and presented different Hub genes and pathways associated with AKI after transplantation. Therefore, we believe that our study, conducted including other GEO datasets, aids in the bioinformatic analysis of AKI development after transplantation. Though the role of hub genes in AKI development has not been fully elucidated, the identified 10-gene biomarkers are thought to be of great help for identifying the development of AKI after kidney transplantation.

## 5. Conclusions

In this study, three GEO datasets with mRNA and miRNA expression information were analyzed to identify DEGs in AKI after transplantation. We then used bioinformatics approaches for the functional enrichment analysis with protein–protein interaction (PPI) network integration. We identified 10 hub genes that may be related to the AKI development after kidney transplantation. These findings may provide biomarkers for diagnostic and therapeutic targets of AKI and suggest their mechanisms. Further experiments and functional studies are still needed to validate the role in DEGs and post-transplant AKI.

## Figures and Tables

**Figure 1 medicina-58-00681-f001:**
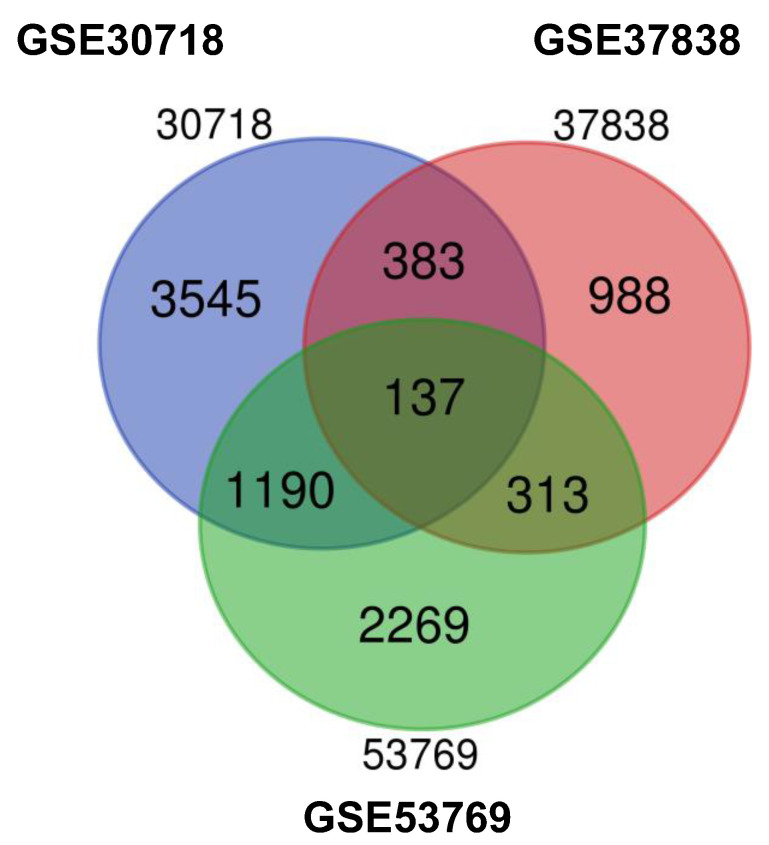
Venn diagram of DEGs common to all three GEO datasets. DEGs, differentially expressed genes; GEO, Gene Expression Omnibus.

**Figure 2 medicina-58-00681-f002:**
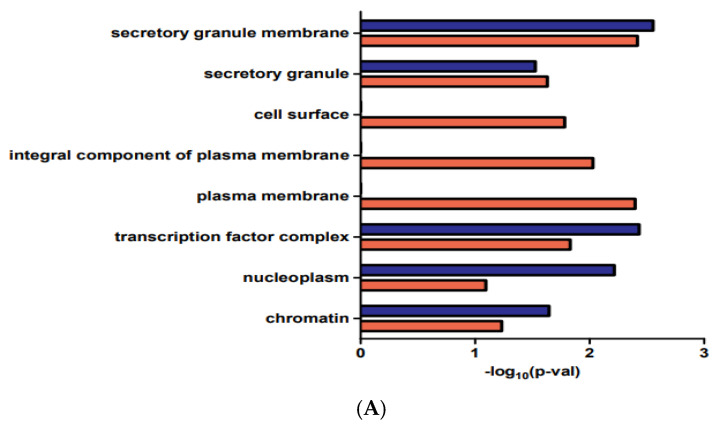
GO and KEGG enrichment analysis of DEGs. (**A**) CC functional classification terms of genes. (**B**) BP functional classification terms of genes. (**C**) MF functional classification terms of genes (**D**) KEGG analysis of genes. BP, biological process; CC, cellular component; DEGs, differentially expressed genes; GO, Gene Ontology; KEGG, Kyoto Encyclopedia of Genes and Genomes; MF, molecular function.

**Figure 3 medicina-58-00681-f003:**
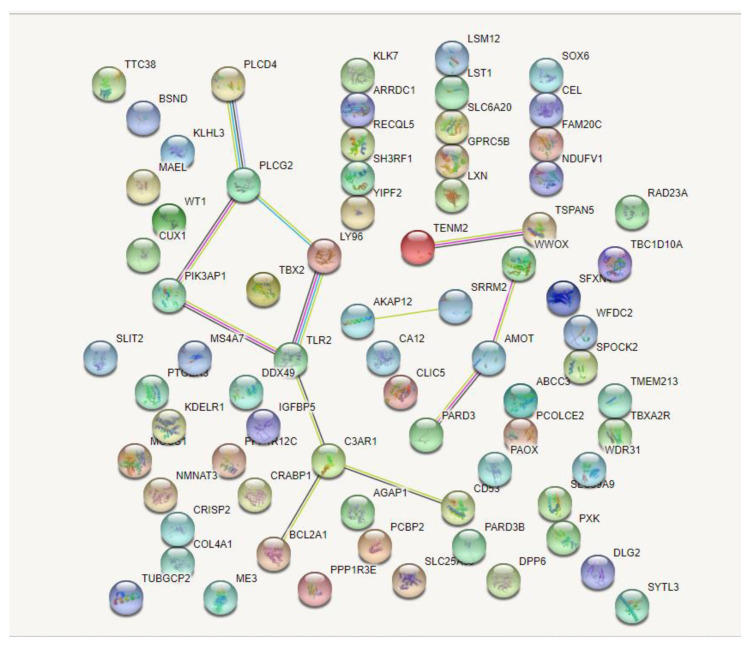
Overall protein–protein interaction network of the 137 identified DEGs.

**Figure 4 medicina-58-00681-f004:**
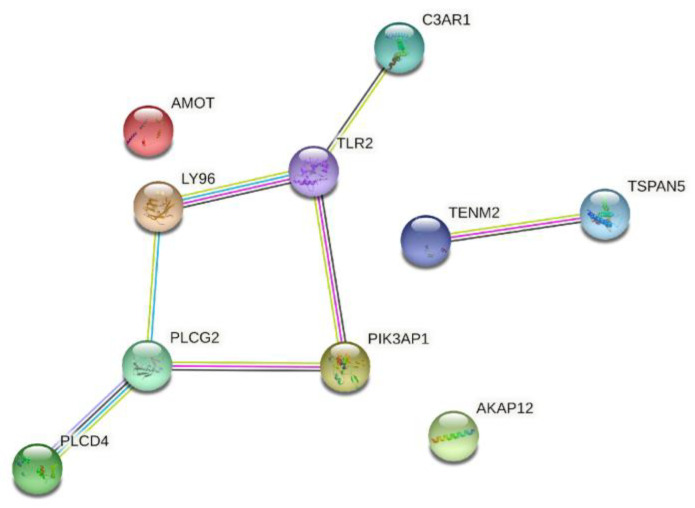
Overall protein–protein interaction network of top 10 genes with the highest connectivity.

**Table 1 medicina-58-00681-t001:** Top 10 hub genes with higher degrees of connectivity.

Gene Symbol	Gene Description	Up/Down
*AKAP12*	A-kinase anchoring protein 12	Up
*AMOT*	angiomotin	Down
*C3AR1*	complement C3a receptor 1	Up
*LY96*	lymphocyte antigen 96	Up
*PIK3AP1*	phosphoinositide-3-kinase adaptor protein 1	Up
*PLCD4*	phospholipase C delta 4	Down
*PLCG2*	phospholipase C gamma 2	Down
*TENM2*	teneurin transmembrane protein 2	Down
*TLR2*	Toll like receptor 2	Up
*TSPAN5*	tetraspanin 5	Down

## Data Availability

Not applicable.

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
