# Peer review of "Identification of Multiple Hub Genes in Acute Kidney Injury after Kidney Transplantation by Bioinformatics Analysis"

_medicina, 2022, doi:10.3390/medicina58050681_

Round 1

Reviewer 1 Report

1) The following phrases should be moved to methods/discussion section, rather than introduction: “Bioinformatics analyses were based on the obtained DEGs. We then used bioinformatics approaches for the functional enrichment analysis with protein–protein interaction (PPI) network integration. This study’s findings may increase our understanding of the mechanism involved in the development of post-transplant AKI”

2) The novelty of the study should be highlighted in the introduction section. There is a previous published article (https://dx.doi.org/10.7717%2Fpeerj.10441), which is similar to some extent, including Gene Expression Omnibus database access, identification of differentially expressed genes (DEGs) and identification of hub genes for PPI network.

3) The aims of the study should be described in the introduction;

4) It is not clear whether up- or down-regulation of the ten hub-genes identified was statistically significant in AKI patients with kidney transplant  

5) The AKI definition used should be specified in methods section. Also, the authors should specify if delayed graft function mentioned in the introduction section was included in analysis

6) Conclusions should be provided after the discussion section  

Author Response

Dear Editors and reviewers,

We thank you for insightful comments and suggestions that you have provided. We have made every effort to address the concerns that were raised, and we feel the paper is stronger for the inclusion and consideration of these points. Below we have reproduced the reviewer comments, following each with a response indicating what changes have been made to the manuscript.

Reviewer 1>

1) The following phrases should be moved to methods/discussion section, rather than introduction: “Bioinformatics analyses were based on the obtained DEGs. We then used bioinformatics approaches for the functional enrichment analysis with protein–protein interaction (PPI) network integration. This study’s findings may increase our understanding of the mechanism involved in the development of post-transplant AKI”

Ans) Thank you for your comments. We moved it to the discussion section.

2) The novelty of the study should be highlighted in the introduction section. There is a previous published article (https://dx.doi.org/10.7717%2Fpeerj.10441), which is similar to some extent, including Gene Expression Omnibus database access, identification of differentially expressed genes (DEGs) and identification of hub genes for PPI network.

Ans) Thank you for your comments. Bioinformatics analysis has recently been widely applied to investigate the mechanisms of various diseases. However, the molecular mechanisms and effects of AKI after kidney transplantation have not been investigated much. And analysis of whole-genomic mRNA and miRNA expression profiles in kidney transplant patients can play an important role in studying the development of AKI after transplantation. So, in this study, we performed an integrative bioinformatics analysis to identify novel gene signatures associated with AKI that occur after kidney transplantation using three data sets with mRNA and miRNA expression information. We added it in introduction section.

3) The aims of the study should be described in the introduction;

Ans) Thank you for your comments. We added the aims of the study in introduction section.

4) It is not clear whether up- or down-regulation of the ten hub-genes identified was statistically significant in AKI patients with kidney transplant

Ans) Thank you for your comments. When we analysis of ten hub-genes, we collect it passed the below criteria. 1) showed statistically significant, 2) trend is also the same via another dataset, and 3) existed in every dataset.

5) The AKI definition used should be specified in methods section. Also, the authors should specify if delayed graft functionn mentioned in the introduction section was included in analysis

Ans) Thank you for your comments. In the Prue AKI dataset, acute rejection and renal disease by histologic criteria, and non-diagnostic histological lesions were excluded. We selected the “pure AKI” datasets with DGF. In the end, we selected three separate gene expression profiles (GSE37838, GSE30718, and GSE53769) based on the GPL570 platform. We revised it.

6) Conclusions should be provided after the discussion section

Ans) Thank you for your comments. We revised the conclusion.

Reviewer 2 Report

I

I read with interest this paper regarding a bioinformatic approach in AKI after KT. However, I have serious concerns about the adopted methodology.

  • Considering the three groups of gene expression profiles (GSE37838, GSE30718, and 85 GSE53769), the number of AKI vs. normal samples is limited, and no information is available about the characteristics of the specific damage. The definition of "AKI" in the transplant setting should be expanded, considering that AKI could be ascribed to many different conditions with entirely different pathogenetic basis (e.g., acute cellular or antibody-mediated rejection; ischemia-reperfusion injury; calcineurin inhibitor toxicity and many more). Donor type characteristics  (living/deceased, expanded or standard-criteria, age) are also deeply involved in DGF and kidney damage and are not even mentioned in the results interpretation.
  • Apart from gene expression analysis, effective upregulation of the involved pathway (i.e., with WB or IF studies) should be considered.

Author Response

Dear Editors and reviewers,

We thank you for insightful comments and suggestions that you have provided. We have made every effort to address the concerns that were raised, and we feel the paper is stronger for the inclusion and consideration of these points. Below we have reproduced the reviewer comments, following each with a response indicating what changes have been made to the manuscript.

I read with interest this paper regarding a bioinformatic approach in AKI after KT. However, I have serious concerns about the adopted methodology.

1) Considering the three groups of gene expression profiles (GSE37838, GSE30718, and 85 GSE53769), the number of AKI vs. normal samples is limited, and no information is available about the characteristics of the specific damage. The definition of "AKI" in the transplant setting should be expanded, considering that AKI could be ascribed to many different conditions with entirely different pathogenetic basis (e.g., acute cellular or antibody-mediated rejection; ischemia-reperfusion injury; calcineurin inhibitor toxicity and many more). Donor type characteristics  (living/deceased, expanded or standard-criteria, age) are also deeply involved in DGF and kidney damage and are not even mentioned in the results interpretation.

Ans) Thank you for your comments. In the Prue AKI dataset, acute rejection and renal disease by histologic criteria, and non-diagnostic histological lesions were excluded. We selected the “pure AKI” datasets with DGF. In the end, we selected three separate gene expression profiles (GSE37838, GSE30718, and GSE53769) based on the GPL570 platform. And the donor type of each dataset has been summarized. We revised it.

2) Apart from gene expression analysis, effective upregulation of the involved pathway (i.e., with WB or IF studies) should be considered.

Ans) Thank you for the great advice. Yes, that is included in our next further study plan. We will validate statistically significant genes via protein expression methods. We added it in discussion section.

Reviewer 3 Report

Congratulations for your work!

I believe it will be suitable to add the limitations of your study.

Conclusion is missing in the text, but present in the abstract. 

I suggest to detail a little bit mor the discussions in comparison with other studies that used similar novel gene markers associated with the development of AKI after kidney (like EBF3, for example)

I suggest you the following articles:

  1. Bi H, Zhang M, Wang J, Long G. 2020. The mRNA landscape profiling reveals potential biomarkers associated
    with acute kidney injury AKI after kidney transplantation. PeerJ 8:e10441 http://doi.org/10.7717/peerj.10441
  2. Xia Zhai, Hongqiang Lou, and Jing Hu. Five-gene signature predicts acute kidney injury in early kidney transplant patients. Aging (Albany NY). 2022 Mar 31; 14(6): 2628–2644.

Author Response

Dear Editors and reviewers,

We thank you for insightful comments and suggestions that you have provided. We have made every effort to address the concerns that were raised, and we feel the paper is stronger for the inclusion and consideration of these points. Below we have reproduced the reviewer comments, following each with a response indicating what changes have been made to the manuscript.

I suggest to detail a little bit mor the discussions in comparison with other studies that used similar novel gene markers associated with the development of AKI after kidney (like EBF3, for example)

I suggest you the following articles:

  1. Bi H, Zhang M, Wang J, Long G. 2020. The mRNA landscape profiling reveals potential biomarkers associated
    with acute kidney injury AKI after kidney transplantation. PeerJ 8:e10441 http://doi.org/10.7717/peerj.10441
  2. Xia Zhai, Hongqiang Lou, and Jing Hu. Five-gene signature predicts acute kidney injury in early kidney transplant patients. Aging (Albany NY). 2022 Mar 31; 14(6): 2628–2644.

Ans) Thank you for your comments. Two other studies showed bioinformatics analysis of AKI development after transplantation. They investigated different GEO datasets and also presented different Hub genes and pathways related to the AKI after transplantation, respectively. Therefore, we believe that our study, conducted including other GEO datasets, aids in bioinformatic analysis of AKI development after transplantation. A comprehensive bioinformatics analysis or some other datasets based on larger sample size is required to validate our findings. And clinical information is also needed to integrate into further study. We added it to the discussion section.

Round 2

Reviewer 1 Report

no further comments

Reviewer 2 Report

The authors performed all suggested revisions. Although some concerns remain about the adopted methodology, the paper could be accepted in this form.